# DLTN-LOSP: A Novel Deep-Linear-Transition-Network-Based Resource Allocation Model with the Logic Overhead Security Protocol for Cloud Systems

**DOI:** 10.3390/s23208448

**Published:** 2023-10-13

**Authors:** Divya Ramachandran, Syed Muhammad Naqi, Ganeshkumar Perumal, Qaisar Abbas

**Affiliations:** 1Department of Information Technology, PSNA College of Engineering and Technology, Dindigul 624622, Tamil Nadu, India; divya07i211@psnacet.edu.in; 2Department of Computer Science, Quaid-i-Azam University, Islamabad 44000, Pakistan; smnaqi@qau.edu.pk; 3College of Computer and Information Sciences, Imam Mohammad Ibn Saud Islamic University (IMSIU), Riyadh 11432, Saudi Arabia; gpperumal@imamu.edu.sa

**Keywords:** cloud computing, resource management, security, deep linear transition network (DLTN), adaptive Mongoose optimization algorithm (AMOA), logic overhead security protocol (LOSP), BAN logic

## Abstract

Cloud organizations now face a challenge in managing the enormous volume of data and various resources in the cloud due to the rapid growth of the virtualized environment with many service users, ranging from small business owners to large corporations. The performance of cloud computing may suffer from ineffective resource management. As a result, resources must be distributed fairly among various stakeholders without sacrificing the organization’s profitability or the satisfaction of its customers. A customer’s request cannot be put on hold indefinitely just because the necessary resources are not available on the board. Therefore, a novel cloud resource allocation model incorporating security management is developed in this paper. Here, the Deep Linear Transition Network (DLTN) mechanism is developed for effectively allocating resources to cloud systems. Then, an Adaptive Mongoose Optimization Algorithm (AMOA) is deployed to compute the beamforming solution for reward prediction, which supports the process of resource allocation. Moreover, the Logic Overhead Security Protocol (LOSP) is implemented to ensure secured resource management in the cloud system, where Burrows–Abadi–Needham (BAN) logic is used to predict the agreement logic. During the results analysis, the performance of the proposed DLTN-LOSP model is validated and compared using different metrics such as makespan, processing time, and utilization rate. For system validation and testing, 100 to 500 resources are used in this study, and the results achieved a make-up of 2.3% and a utilization rate of 13 percent. Moreover, the obtained results confirm the superiority of the proposed framework, with better performance outcomes.

## 1. Introduction

In present times, cloud computing [1,2] has become a major trend in many areas of our lives, including businesses, government, entrepreneurship, manufacturers, and business. A parallel and distributed system known as cloud computing allows for the sharing of centralized data storage, data processing tasks, and online access to information technology services or resources by connecting groups of remote servers. When using cloud computing [3,4,5], the user leases the resource rather than acquiring it, which lowers the cost of the software. Additionally, cloud computing offers on-demand services that clients can access from faraway locations at any time. With cloud computing, it is possible to use a remote computer’s services instead of storing and retrieving data from the local computer. Clients use cloud services rather than maintaining their own infrastructure, so users do not need to be familiar with network infrastructure. Resource provisioning [6,7] in real time has emerged as one of the primary challenges in modern highly distributed systems, and it is among the key problems in cloud computing.

The cloud offers clients virtualized resources [8,9] as a service over the Internet in all circumstances, with high quality, in order to maximize network utility. The virtualized architecture enables resource re-allocation by transferring VMs between hosts. Moreover, the virtualized resources are mapped to physical resources to accomplish resource provisioning [10,11]. Significant advantages of resource re-allocation in distributed systems include distributing and balancing the load on processors, leveraging the bandwidth of the network, using the facilities of cloud data centers under differing workloads, and reducing the overall request execution time. Two steps to sharing resources are scheduling and mapping. Planning allows for the location of resources for network hosts and connections. There are many resources used in the data center [12]. These computational resources are provided by a third party known as “cloud providers”, who oversee giving customers access to resources whenever they require them.

Virtualization is a process that obscures away all of the physical resources [13,14]. Moreover, it is a platform setup to share resources and improve the flexibility of data sharing. With virtualized resources that the virtualization monitors or hypervisor creates, applications run on the host operating system. Cloud task scheduling [15,16] is the process of deciding a task’s start time within a workflow while considering all of the task’s dependencies and expiration dates. For independent tasks, the task scheduling procedure is typically not challenging [17]. However, a workflow’s task dependencies can make scheduling a difficult issue. The right resources should be allotted to the tasks after scheduling and prioritizing them. Depending on how long it will take to complete each task, the appropriate processing resource is needed. Inefficient resource provisioning not only lengthens the workflow’s execution time, increases the cost, and uses more energy, but it also decreases the effectiveness of resource usage. However, if load balancing is neglected, the resources’ efficiency can be severely reduced. As a result, cloud service providers [18,19] have significant concerns regarding task scheduling and resource provisioning. The scheduling process for these workflows is also more difficult to manage than it is for most other workflow types because of the task dependencies. In the case of online workflows, this complexity may even increase. When planning tasks and allocating resources in cloud environments, a number of goals [20,21] can be taken into account, such as lowering energy consumption, load balancing, and increasing resource utilization. In the literature, several resource management strategies have been put forth, each of which aims to accomplish one or more goals in the most effective way possible. Different learning-based techniques have been used to improve the efficacy and efficiency of cloud resource management [22,23]. However, the existing techniques limit the following problems: high processing time, delay, ineffective resource allocation, and lack of reliability. Therefore, the proposed work intends to develop a new resource allocation and integrated security framework for cloud systems.

### 1.1. Major Contributions

The key contributions of the proposed work are as follows:To effectively allocate the resources in the cloud system with reduced makespan and overall processing time, a Deep Linear Transition Network (DLTN) model is developed.An Adaptive Mongoose Optimization Algorithm (AMOA) is used to figure out the beamforming solution for reward prediction while allocating resources to the cloud systems.To ensure the security of resource management, an advanced Logic Overhead Security Protocol (LOSP) is deployed, which works based on Burrows–Abadi–Needham (BAN) logic.To validate the performance of the proposed DLTN-LOSP framework, an extensive performance and comparative analysis was conducted during the evaluation.

### 1.2. Paper Organization

The following sections encompass the remaining parts of this article: A detailed review of the cloud resource allocation and security models is presented in Section 2, which also discusses the benefits and drawbacks of each model, considering their performance results. The proposed DLTN-LOSP framework is clearly described in Section 3, along with an explanation of the overall workflow and algorithms. Additionally, Section 4 compares and validates the effectiveness of the proposed resource allocation and security model based on various metrics. Section 5 includes discussions of this study’s limitations and future works. The overall paper is concluded in Section 6 with a discussion of its future scope, conclusions, and implications.

## 2. Literature Review

A literature review of previous works on cloud resource management and security management is presented in this section. It also examines each mechanism’s benefits and drawbacks regarding scheduling and security functions.

Hui et al. [24] deployed a secure new resource allocation mechanism for mobile edge computing systems. The main purpose of this paper was to ensure the stability and safety of mobile edge systems by properly allocating user resources. Here, differential equation modeling, termed Lyapunov stability theory, was used to enable efficient resource allocation. Resource allocation and other issues in communication networks could be resolved in this work using this approach. Meng et al. [25] created a dynamic scheduling mechanism that takes security into account and uses the standard particle swarm optimization (PSO) algorithm to distribute cloud resources in industrial applications. The goal of resource allocation in the scheduling problem is to optimize resource deployment in each application task to meet the various requirements of users. Moreover, the dynamic workflow model is also developed to balance the performance of both security and resource scheduling. Here, each cloud resource and task of an industrial application is assigned a security degree, which indicates how well it performs in terms of security. However, the performance of the suggested mechanism was not up to par, which could be the major limitation of this work.

Abid et al. [26] investigated the different types of challenges and issues correlated to cloud resource allocation. Typically, resource allocation in the cloud is defined as the process of allocating the set of obtainable resources to the applications with an ensured service level agreement (SLA), reduced cost, and reduced energy. Every user of a cloud service desires the maximum number of resources for a particular task that must be completed on time and thus can improve performance. Here, some of the major problems associated with secure cloud resource allocation are also discussed, as shown below: Cloud service providers find it very challenging to predict the needs of their customers’ applications while also accommodating their desire for on-time task completion. Every virtual machine that runs on a physical machine should be able to meet its resource requirements, but consumers also require networking services with effective quality of service (QoS) to ensure the efficient delivery of their application data. If a job is expected to take longer than usual, the service provider needs to schedule the resources’ availability. As a result, a method is required that can manage the interruption and switch the task to the available resource. Due to rising energy prices and the need to reduce greenhouse gas emissions while also lowering overall energy consumption, communications, and storage, energy-efficient resource allocation is one of the unsolved challenges in cloud computing. Moreover, the cloud resource allocation mechanisms are categorized into the following types: dynamic resource allocation and AI-based resource scheduling. When compared to the dynamic models, the AI-based resource allocation mechanism has the major advantages of reduced cost, energy, execution time, and workload.

Table 1 summarizes the literature review with the methodology, results, and limitations of each work. Shukur et al. [27] highlighted some of the recent methods and scheduling approaches used for resource allocation in cloud computing via data center virtualization. The goal of the paper is to examine how effectively providing resources based on client requirements could be managed through virtualization. The outcomes of these methods demonstrated that every suggested method and scheduling algorithm can utilize the shared resources of the cloud data center. Du et al. [28] implemented a deep reinforcement learning (DRL) algorithm for optimally allocating resources in the dynamic cloud system. Here, the DRL was developed by incorporating the functions of the long short-term memory (LSTM) model with the fully connected neural network (NN) technique. The cloud provider chooses the server with the capacity to host each virtual machine (VM) request and the cost of running the VM on the chosen server, both of which are associated with profit maximization. Moreover, the classifier training and testing operations are performed using the samples obtained from the Azure dataset. The key benefit of this approach is that the outcomes are quite encouraging under various user arrival patterns, and it successfully accommodates more user requests with increased profit. Yet, it has major problems such as increased time consumption and overfitting. Naha et al. [29] deployed a deadline-based dynamic resource allocation mechanism for fog–cloud environments. The purpose of this work was to perform resource allocation based on resource ranking in a hierarchical manner. Here, the cloud fog servers are used if the available fog servers are insufficient to complete the given task. Typically, factors such as processing power, available bandwidth, and response time must be considered for the proper selection of cloud resources. Additionally, resource provisioning involves prioritizing the applications in accordance with the volume of requests received from the environment. The primary advantages of this work were reduced processing time and cost. However, it is required to improve the capability of handling resource failure to solve complex situations.

Afrin et al. [30] implemented a multi-objective resource allocation mechanism for a cloud-based smart factory application. Here, the simultaneous optimization of time, energy use, and cost is considered while allocating resources for the tasks of a robotic workflow. Moreover, a multi-objective evolutionary approach called the NSGA-II algorithm is used to solve the constrained optimization problem for resource allocation. Different types of constraints, such as assignment, energy, budget, and data dependency, have been considered. By using an effective optimization technique, it efficiently handled the process of resource allocation in cloud systems. Nevertheless, it has the problems of high cost and time, which affect the performance of the entire framework. Haji et al. [31] investigated the different types of strategies used for dynamic resource management. The purpose of this work was to improve the quality of service (QoS) parameters by properly allocating user tasks on cloud systems.

Typically, improving communication bandwidth, reducing traffic, and reducing storage complexity are the most essential factors that need to be accomplished in a successful resource management system. Thein et al. [32] constructed a new framework, termed an energy-efficient resource allocation model for cloud systems. This paper intends to reduce the cost of scheduling and increase the revenue rate by properly scheduling the resources across the cloud systems. Moreover, it uses fuzzy logic and reinforcement learning mechanisms to obtain an environmentally friendly solution to the resource allocation problem. Typically, virtualization aims to conceal physical and low-level system components from users. As a result, resources can be configured and used efficiently, allowing for the hosting and concurrent operation of multiple applications. This method performs multidimensional mapping that complies with a set of rules, such as the number of available CPU cores, the amount of needed storage hard disk space, the amount of physical memory, the allocated time, and network bandwidth. The advantages of this work were guaranteed service level agreements (SLA), efficient resource utilization, and energy-efficient allocation. Praveenchandar et al. [33] deployed a dynamic resource allocation mechanism with an enhanced power management strategy for cloud systems. Moreover, the preference-based task-scheduling (PBTS) mechanism incorporated with the dynamic resource table (DRT) was used in this work for dynamically scheduling user tasks on cloud machines. The steps in this task schedule are as follows:(1)The PBTS algorithm process.(2)Updating the resource table based on the connectivity established by an agent. Typically, each data center’s energy usage is directly correlated with resource exploitation. In addition, the power management algorithm (PMA) was used to save the energy of resources, which minimizes the time delay and response time.

As a result of this review, it was discovered that different approaches to resource allocation and security in cloud systems have been developed in the past. However, they suffer from the following serious issues:○Increased make-up and processing time.○Delay in the process.○High power consumption.○Lack of system efficacy and reliability.○Reduced QoS.

Yiqiu et al. [34] presented a task-scheduling algorithm based on load balancing and cloud computing. These algorithms can lead to resource underutilization or overutilization and may be sensitive to changes in workload patterns. Additionally, optimizing one performance metric may negatively impact another. Finally, evaluating the effectiveness of load-balancing algorithms can be time-consuming and resource-intensive.

Therefore, the proposed work intends to develop a new optimization-integrated deep learning model for ensuring proper resource allocation and security in cloud systems.

## 3. Materials and Methods

This section presents a clear explanation of the proposed deep-learning-based resource allocation and security model for cloud systems. The original contribution of this work is to develop a novel framework for allocating resources in the cloud system with guaranteed security. In this paper, algorithms such as the Deep Linear Transition Network (DLTN), the AMOA, and the LOSP are used together. After modeling the system, the signal-to-interference-plus-noise ratio (SINR) is estimated, and the empirical power model is first constructed. Then, the DLTN algorithm is implemented to decide on resource allocation, where the random action selection, convolutional feature estimation, model training, and loss function computation operations have been performed. Consequently, the beamforming solution is optimally computed with the help of the AMOA, which is used to predict the reward for resource allocation. After resource allocation, security is ensured for the allocation system by using the LOSP, where BAN logic is used to predict the agreement logic. The primary objectives are as follows:(a)To create a dynamic process for task scheduling and resource allocation.(b)To maximize throughput through optimal scheduling while reducing the anticipated total span.

The overall workflow model of the proposed framework is shown in Figure 1, which comprises the following operations:DLTN.AMOA.LOSP.

**Figure 1 sensors-23-08448-f001:**
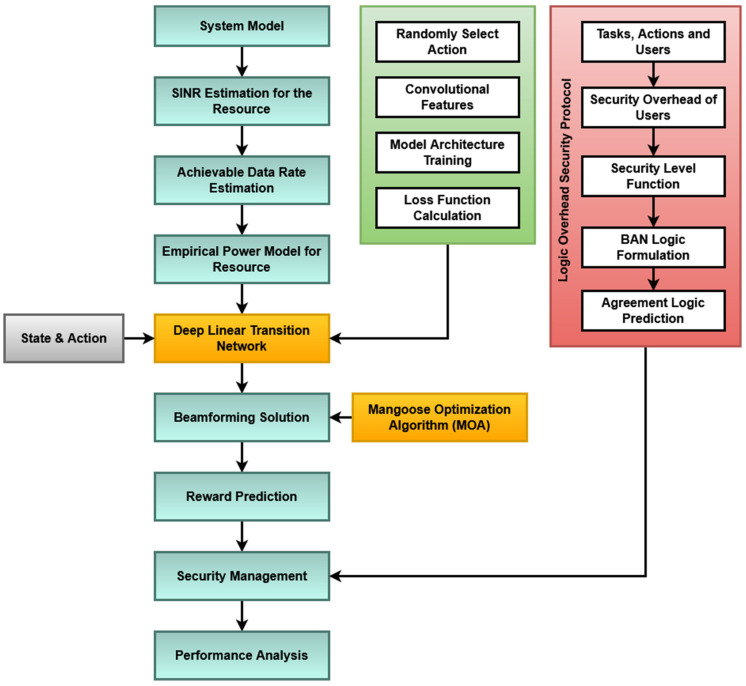
Workflow model of the proposed framework.

A. Deep Linear Transition Network (DLTN): In this framework, the system is modeled with a set of cloud users and tasks at first. Here, the service provider receives a request from the cloud users to access various cloud resources. The DLTN algorithm is presented below as Algorithm 1 and it is mainly used to estimate the cumulative reward for allocating user tasks to the cloud machines. It is a kind of deep learning algorithm mainly used to make appropriate decisions for task scheduling and allocation. Typically, various machine-learning-based decision-making algorithms are implemented in the existing work for allocating user tasks to the cloud machines for execution. In order to improve scheduling efficiency and performance, the DLTN algorithm is implemented in this work. Here, the SINR is estimated for the resource by using the following model:(1)INRŰ=guHѿu2∑r≠ŰguHѿu2+ϑ2
where ѿu is a binary indicator that denotes whether the ϑth resource blocks of base station are allocated to user u, and guH is the channel gain between base station and user u. After that, the data rate between the base station and user is estimated by using the following model:(2)Dk,uR=∑f∈FBflog2⁡(1+ϑuk)
where Dk,uR indicates the data rate, F denotes the set of all the RBs that can be allocated, and Bf is the bandwidth for the particular resources. Consequently, the total delay for the user u is computed, which comprises three components such as transmission delay, queuing delay, and retransmission delay, and is estimated as shown below:(3)τutotal=τutx+τuQ+τurx
where τutx represents the transmission delay, τuQ denotes the queuing delay, and τurx indicates the retransmission delay. Then, the transmission delay is estimated by using the following model:(4)τutx=Dk,uRCk,u
where Ck,u is the capacity for the user u with respect to k bandwidth. In this model, the optimization process is used to attain maximum throughput with reduced delay of processing. Moreover, the state of power control model includes the factors of queue length of data rate, current delay, and transmission power, as computed below:(5)pk,ts={Qm,∑u∈Umτu,Pk|∀u∈U,k∈K}
where Qm indicates the queue length of packets, and Pk denotes the transmission power. To achieve the desired system performance, the agent can adjust the transmission power in accordance with the state definition, which includes the traffic demand and network status. After that, the transmission power is estimated by using the following model:(6)Pk,t=ak,tcPmLm
where ak,tc indicates the action of power control is to choose power level = {1,2,…,Lm}, and Lm denotes the highest power level. Then, the reward of power control is estimated according to the weighted sum reward with the penalty of large transmission power, as shown in below:(7)rk,te=∑m∈Mkwmrme−φαk,t
where wm and rme denote the priority weight and reward of each slice, respectively, αk,t is the random number that varies between {0,1} ∀k,t, and φ is a constant. The expected long-term reward is maximized as shown in below:(8)Dp,a=Erte+βDpt+1,at+1pt=p,at=a
**Algorithm 1** Deep Linear Transition Network (DLTN)Input: Cloud RAN ƦRRH, set of users Ű, action decision ãnOutput: Cumulative reward ɸProcedure:Step 1: signal-to-interference-plus-noise ratio (SINR) at the receiver of user Ű={1,2,…,U} can be estimated by using Equation (1).Step 2: data rate between BS k and user u is denoted as Dk,uR, and it can be calculated using Equation (2).Step 3: the total delay of user u is computed using Equation (3), which consists of three components, the transmission delay τutx, queuing delay τuQ, and retransmission delay τurx.Step 4: the transmission delay is formulated as shown in Equation (4).Step 5: the state of power control model includes the queue length of data rate, current delay, and current transmission power, which is estimated using Equation (6).Step 6: Action: the transmission power is estimated with respect to the action of power control by using Equation (7).Step 7: Reward: the reward of power control model is computed with the weighted sum reward and penalty of large transmission power as shown in Equation (8).Step 8: the expected long-term reward is maximized by using Equation (9).Step 9: then, the learning rate and discount factors are computed using Equation (9).Step 10: finally, the cumulative reward is produced as the output, which is used for allocating the resources in cloud systems.

The parameter update in DLTN is performed as represented in the following model:(9)δt+1=δt+φ[rte+βmaxp′Dpt+1,at+1;δt−Dpt,at;δt]∇Dpt,at;δt
where δ, p, and β, respectively, denote the parameters of DLNN, the learning rate, and the discount factor. By using this model, the cumulative reward is computed, which can be further used for reward prediction.

B. Adaptive Mongoose Optimization Algorithm (AMOA): For improving the process of resource allocation, the beamforming solution is computed in this framework, which is performed by using an AMOA. Generally, different types of optimization techniques are implemented in the existing work for performing resource allocation in cloud systems. The key benefits of using this approach are as follows: increased convergence rate, reduced iteration count, and minimal processing time. In this model, the population is stochastically generated with upper and lower bounds, as shown below:(10)$$X=\left(\begin{array}{lllll} x_{1,1} & x_{1,2} & \ldots & x_{1,d-1} & x_{1,d}\\ x_{2,1} & x_{2,2} & \ldots & x_{2,d-1} & x_{2,d}\\ & & x_{a,b} & & \\ x_{m,1} & x_{m,2} & \ldots & x_{m,d-1} & x_{m,d} \end{array}\right)$$
where X represents the set of the candidates’ present population that is randomly generated, $x_{a,b}$ indicates the position of the $b^{th}$ dimension of the $a^{th}$ population, m is the population size, and d indicates the problem dimension. Then, the best solution at each iteration is estimated according to the following model:(11)$$x_{a,b}=unifrnd(V_{Min},V_{Max},V_{size})$$
where unifrnd denotes the random number that is uniformly distributed, $V_{Min}\;\mathrm{and}\;V_{Max}$ are lower bound and upper bound, respectively, and $V_{size}$ is the problem dimension. Here, the alpha female ($\alpha^{f}$) is considered as the family unit controller that is selected based on the following model:(12)$$\alpha^{f}=\frac{F_{j}}{\sum_{j=1}^{m}F_{j}}$$
where $m-b^{s}$ matches the number of mongooses in the alpha group, $b^{s}$ is the number of babysitters, and $peep^{x}$ represents the sound of female alpha to the path of the other unit members. Then, the sleeping mound is determined by abundant food, as represented below:(13)$$X_{j+1}=X_{j}+\delta \times peep^{x}$$
where δ is a random uniformly distributed number [−1, 1] generated after each iteration. Below is an evaluation of the sleeping mound:(14)$$sl_{j}^{m}=\frac{F_{j+1}-F_{j}}{\underset{j\to 1\mathrm{to}m}{\max}\left\{|F_{j+1},F_{j}|\right\}}$$

The following equation is used to calculate an average value when a sleeping mound is discovered:(15)$$\rho =\frac{\sum_{j=1}^{m}sl_{j}^{m}}{m}$$

The next stage is scouting, which assesses the next sleeping mound determined by another food source, after the babysitter exchange criterion is met. Typically, the mongoose is known to forage and scout at the same time, with the reasoning that the further the unit forages, the more likely it is to find the next sleeping mound.
(16)$$X_{j+1}=\left\{\begin{array}{c} X_{j}-c^{v}\times \delta \times rand\times [X_{j}-\vec{M_{m}}\;if\;\rho_{j+1}>\rho_{j}]\\ X_{j}+c^{v}\times \delta \times rand\times [X_{j}-\vec{M_{m}}\;else] \end{array}\right.$$
(17)$$c^{v}={\left(1-\frac{i^{r}}{M_{i^{r}}}\right)}^{\left(2*\frac{i^{r}}{M_{i^{r}}}\right)}$$

Here, rand is a random number between [0, 1], and $c^{v}$ is a parameter that controls the group’s collective, volatile movement and linearly decreases over iterations.
(18)$$\vec{M_{m}}=\sum_{j=1}^{m}\frac{X_{j}*sl_{j}^{m}}{X_{j}}$$

Here, $\vec{M_{m}}$ indicates the force that propels mongooses to move to a new sleeping mound.

C. Logic Overhead Security Protocol (LOSP): For effective security management, the LOSP is deployed in the proposed work, which ensures secured resource allocation in the cloud system. This security protocol is developed based on BAN logic, which is a popular and emerging security verification model. Moreover, it is extensively used in many security applications for performing key authentication and session key agreement operations. The following are the definitions and notations used in BAN logic:
○The protocol’s agents are referred to as $P_{r}^{l}$.○The messages are symmetrically encrypted using $K_{e}^{y}$.○Keys and $P_{e}^{K}$ are comparable, with the exception that is employed in pairs.○$N_{e}^{c}$ stands for message components that should not be repeated.○Since they are not likely to be repeated, $T_{t}^{s}$ are analogous to nonces.○The relevant BAN logic assertions listed below are useful for examining the security of the proposed protocol:○B1:Q|≡Y:Q would be justified in believing Y if Q believes it. Q may assume that Y is true in particular.○B2:Q<Y:Q sees Y. Q is able to read and repeat some messages from Y that it has received.○B3:Q|~Y:Q once said Y. At some point, Q sent a message that contained the sentence Y. Whether this is a replay is unknown. It is known that Q trusted Y when it was sent, though.○B4:Q⇒Y:Y is subject to Q’s authority. The primary Q is an expert on X and can be relied upon in this regard.○B5:♯(Y): Y is a new message.○B6:(Y,Z): one component of the formulas (Y, Z) is the specifications Y or Z.○$B7:<Y{>}_{Z}:$ the formula Y plus formula Z combined.○B8:{Y}M: using the formula M, the function Y is encrypted.○B9:(Y)M: the key M is used to hash the expression Y.○B10:QM↔C: primary Q and C communicate using the common key M.○B11:Q⇔C: only Q and C are privy to the equation Y, and only a principal they can trust will be able to use it.

In this work, the following BAN logical postulates are pertinent. What follows is the message-meaning rule:(19)QM↔C,Q←CQ|≡C|~Y

If the principal Q assumes the secret key M is shared with the principal C and Q receives message *Y* that has been encrypted with M, Q then thinks that message Y was once sent by the principal C. The rule of conjunctive freshness is as follows:(20)$$\frac{Q|\equiv \neq (Y)}{Q|\equiv \neq (Y,Z)}$$

If the principal considers that Y is fresh, the principal Q then thinks that (Y,Z) is fresh. The belief rule is as follows:(21)$$\frac{Q|\equiv \left(Y\right),Q|\equiv (Z)}{Q|\equiv \neq (Y,Z)}$$

If the principal Q believes Y and Z, the principal Q then believes (Y,Z).

The nonce verification rule is as follows:(22)$$\frac{Q|\equiv \neq \left(Y\right),Q|\equiv C|\sim Y}{Q|\equiv C|\equiv Y}$$

If the principal Q considers Y to be new and the principal C already sent Y, the principal Q then considers that C has Y. The jurisdiction rule is as follows:(23)$$\frac{Q|\equiv C\to Y,\;Q|\equiv C|\equiv Y}{Q\;|\equiv Y}$$

If both the principal and C concur that C has jurisdiction over Y, Q then concurs that Y is true. The session key rule is as follows:(24)Q|≡≠(Y), Q|≡C|≡YQ|≡Q M↔ C

The principal Q considers that the user shares the session key M with C if principals Q and C believe that Y is a required parameter of the session key, and principal Q believes that the session key is new.

By using this model, the secured resource scheduling and management are ensured in the proposed framework.

## 4. Results

The suggested methodology presents a cutting-edge framework made to deal with the difficulties of allocating resources in cloud systems while putting security first. The Deep Linear Transition Network (DLTN), Adaptive Mongoose Optimization Algorithm (AMOA), and Logic Overhead Security Protocol (LOSP) are just a few of the cutting-edge techniques used to accomplish this. This strategy is particularly novel because it works to secure the allocation system as well as optimize resource allocation. In the first step of the process, the cloud system is modeled, and important variables such as the signal-to-interference-plus-noise ratio (SINR) are estimated. Following the construction of an empirical power model using these projections, the SINR estimation is crucial because it offers insights into the quality of the resources that are currently accessible, which is a crucial consideration in decisions about resource allocation. A key element of the process is that the DLTN is in charge of choosing how to allocate resources. It requires a number of steps, including random action selection, model training, loss function computation, and convolutional feature estimation. Within the cloud system, decisions on how to allocate resources are influenced by the results of these processes. The methodology uses the Adaptive Mongoose Optimization Algorithm (AMOA) to compute the beamforming solution optimally after judgments about resource allocation have been made by the DLTN. Because it forecasts the reward for resource allocation and helps optimize the allocation process, the AMOA plays a significant role in this situation. This step ensures that resources are allocated efficiently to maximize system performance. Security is a paramount concern in cloud systems, and the proposed methodology addresses this by implementing the Logic Overhead Security Protocol (LOSP). The LOSP employs BAN logic to predict agreement logic, enhancing the overall security of the cloud resource allocation system. This ensures that the allocation process remains secure and resistant to potential threats or vulnerabilities.

### 4.1. Resources Allocation

The allocation of resources should be in line with decentralization, since the provider may supply services of unique sorts or combinations of these services depending on resources, complicating the issue rather than necessitating complicated tenders. Access to pertinent resources is possible from a variety of sources, and multiple users can compete for the same resources by requesting submissions from customers, providers, and suppliers. The majority of problems in cloud computing are related to work planning, increasing memory management, availability of services, control of power, and data security. In the proposed work, efficient resource allocation is carried out with the use of DLTN-LOPS with AMOA mechanisms by optimizing the allocation process. Resources are used to build a layer that boosts the effectiveness of the cloud system. Another element is lowering the cost of using already-existing resources. Here, the time, cost, and fitness parameters are taken into account for efficient resource allocation in the cloud. A cloud data center is constructed and continuously monitored in this system. Additionally, it begins building data centers using resource agents. Initially, each data center had a number of data hosts and related VMs. Then, optimal resource allocation was performed with the use of the proposed DLTN-AMOA techniques.

### 4.2. Experimental Setup

The implementation of the proposed DLTN-LOSP system was carried out using the Python programming language. In DL algorithms, parameters hold significant importance, especially the number of hyperparameters and the total number of iterations, as they greatly influence the algorithm’s performance. Consequently, our experiment encompassed various experiments to identify the most suitable hyperparameters and a number of epochs. To assess the effectiveness of our approach, we compared the performance of the proposed DLTN-LOSP with state-of-the-art techniques. Initially, the code was implemented and tested on a system featuring a Core i7 processor clocked at 3.778 GHz with 32 GB of RAM. The simulation and experimental specifications are illustrated in Table 2.

When choosing a cloud service provider, there are many factors to consider, including programming languages, deep learning frameworks, simulation tools, network equipment, security protocols, resource allocation algorithms, datasets, operating systems, data storage, and monitoring and management tools. There are several cloud service providers to choose from. MyCloud Solutions is used here. The Python 3.8 programming language is also used. The deep learning framework TorchFlow also impacts the decision. For simulation, SimuCloud Pro is used to test the applications and infrastructure before deployment. Network equipment, such as routers and load balancers, from CloudRouter is used.

The SafeNet BAN Suite security protocol is used in the cloud environment. Resource allocation is performed via an adaptive resource allocator (ARA) to optimize the cloud resources. AzureML Sample Data, a popular dataset from Microsoft (https://azure.microsoft.com/en-in/products/open-datasets (accessed on 12 February 2023)) is used. For further research and real-time implementation, data storage is a critical consideration for any cloud-based application. CloudDB Pro, CloudStore S3, and CloudFS X are just a few of the options available. Finally, monitoring and management tools such as CloudWatch Pro and CloudControl Center can help keep track of cloud resources and make informed decisions.

### 4.3. Results Analysis

This section uses a variety of metrics to verify the effectiveness and outcomes of the suggested cloud resource allocation + security framework, termed as DLTN-LOSP. In order to demonstrate the superiority of the proposed framework, the obtained results are also contrasted with some of the baseline approaches. Figure 2 compares the makespan of existing [33] and proposed resource scheduling techniques with respect to varying task sizes (kb). Makespan measures the amount of time needed to complete a set of tasks from beginning to end. Let ${Task}_{Com}$ be the total amount of time needed to complete all tasks, and then represent span as follows:(25)$$Makespan=Maximum({Task}_{Com})$$

This comparative analysis graph shows the decreased makespan of the proposed DLTN-LOSP model, where the efficiency is increased by about 30% when compared to the other algorithms. The simulation graph of the proposed model results in a shorter makespan than the other algorithms.

The duration from the arrival of a task to its successful completion is known as the response time of the task. The response time ${Task}_{Res}$ can be represented as follows: (26)$${Task}_{Res}={Task}_{Com}+{Task}_{Ar}$$
where ${Task}_{Com}$ indicates the task completion time, and ${Task}_{Ar}$ is the task arrival time. When compared to other algorithms, the response time of the proposed DLTN-LOSP model is highly minimized as visually displayed in Figure 3. Consequently, the execution time analysis of the proposed model is illustrated in Table 2.

Table 3 presents a comprehensive time analysis of the proposed DLTN-LOSP model, shedding light on the execution times for various critical components of the system. Notably, task monitoring and scheduling demand the most significant time investment, taking approximately 18 min. This phase encompasses the meticulous tracking and management of tasks within the cloud environment, ensuring efficient allocation and execution. Workload prediction at 9 min plays a pivotal role in forecasting future workloads and resource requirements, allowing proactive resource allocation. In contrast, components such as agent connection (0.048 s) and user response (0.010 s) demonstrate exceptional speed, highlighting the model’s rapid communication and responsiveness to user interactions. Lastly, power management, which takes 1.198 s, contributes to energy efficiency by efficiently managing power resources within the cloud infrastructure. This detailed time analysis not only underscores the temporal intricacies of each model component but also showcases the model’s capacity to deliver efficient and responsive cloud resource allocation while addressing security concerns.

The resource utilization rate ${Res}_{Uti}$ is defined as the quantity of resources made available that are used by the tasks to be completed. It is computed as follows:(27)$${Res}_{Uti}={Res}_{Avai}-{Res}_{Unres}$$
where ${Res}_{Avai}$ is the available resources, and ${Res}_{Unres}$ is the unused resources. The proposed work’s resource usage includes both CPU and memory usage. When compared to the other two algorithms, the proposed algorithm’s utilization rate is consistently higher. As shown in Figure 4, the resource utilization rate is computed for the existing and proposed techniques with respect to varying task sizes (kb). The estimated results indicate that the proposed DLTN-LOSP model provides an improved resource utilization rate when compared to the other models. Similarly, Figure 5 compares the task completion ratio of the existing and proposed techniques with respect to varying task sizes (kb). This analysis also indicates that the proposed DLTN-LOSP reaches the maximum task completion ratio when compared to the other approaches. Due to the inclusion of the reward prediction process, the resources are properly allocated in the cloud systems, which helps to obtain an increased task completion ratio.

Figure 6 compares the total energy consumption of the existing [35] and proposed scheduling mechanisms with respect to the varying size of user tasks (kb). In order to reduce energy consumption, a power management module is used in the proposed work. According to the analysis, it is determined that the total energy consumption of the proposed DLTN-LOSP model is highly reduced when compared to the other approaches. For cloud users, it is crucial to guarantee both service reliability and the quality of service that users have requested. Different service levels can be offered by cloud service providers based on the demands of cloud users. Therefore, users must ensure that they receive services with a guaranteed level of quality based on the price they expect to be paid. Between cloud users and service providers, a preliminary agreement built on quantitative and qualitative standards is established. The service provider will make sure that the level of service is not diminished while resources are being used. As shown in Figure 7, the overall processing time of the existing and proposed scheduling techniques is compared with respect to different test scenarios. Based on the results, it is evident that the proposed method takes less time than the other methods in every scenario. This time is significantly reduced compared to the other methods, especially as the workflow’s node count rises.

Figure 8 compares the total number of allocated VMs for the different test scenarios using the existing and proposed scheduling approaches. This analysis demonstrates how the suggested approach uses less hardware or virtual machines. This will increase resource utilization significantly and lead to more efficient use of resources. The primary cause of the decrease in resources is the use of the reward function, which makes better resource management result in more immediate rewards. This may result in a more effective use of resources, which could lower the total amount needed. Moreover, the average resource utilization rate of existing and proposed scheduling approaches is compared with respect to different test scenarios, as shown in Figure 9. The average resource utilization is higher when the suggested method is used compared to the other methods. The remaining capacity of the resources is taken into account after resource allocation. In fact, a resource will benefit more if a task can be completed on it so that less of its remaining capacity is used up. The result is optimal resource utilization, and every resource tries to follow this pattern. In other words, it is critical to consider how resources are used on average to complete all tasks.

As a result, in addition to the advantages of each resource, the long-term advantages of the entire system are also taken into account. The average use of all resources will increase as a result. The proposed method’s load-balancing performance is validated using the standard as shown in Figure 10. The results determine that the proposed method significantly improves load balancing when compared to the other methods. Resource efficiency cannot be determined solely by average resource utilization. Although resource usage is generally appropriate, some resources might be overloaded while others might be underloaded. Resource availability and efficiency are lowered as a result. A better load balance is achieved with a similar average to the previous one and fewer overloaded and underloaded resources by lowering the standard deviation parameter.

Figure 11 and Figure 12 validate the training and testing accuracy of the existing [36] and proposed scheduling techniques with respect to different numbers of VMs. The main conclusion to be drawn from this graph is that there is a slight reduction in accuracy as the number of deployed VMs rises. This can be stated by the possibility that a greater selection of VMs might make it more likely that some inappropriate VMs will be mistakenly given tasks. When compared to the other deep learning models, the proposed DLTN-LOSP technique provides increased training and testing accuracy due to proper resource allocation.

Figure 13 validates the total utilization rate of existing and proposed scheduling approaches with respect to varying numbers of tasks. In this set of experiments, the quantity of deployed virtual machines (VMs) is changed from 10 to 50 and the quantity of tasks from 100 to 1000. The goal is to investigate the scalability of various solutions with variations in the number of VMs and tasks. The graph demonstrates how the cost of utilization for the various approaches increases as the number of VMs increases. This is a result of increasing resource consumption brought on by the deployment of more VMs. The overall reward (cost) of the present and suggested mechanisms for different numbers of user tasks are depicted in Figure 14 and Figure 15. The observed results show that the DLTN-LOSP technique performs better than other strategies overall.

## 5. Discussions

The assessment comes to a close by identifying typical issues with current resource allocation strategies, including lengthened life spans, prolonged processing times, power consumption, a lack of system effectiveness, decreased QoS, and more. In order to address these issues and guarantee effective resource allocation and security in cloud systems, the proposed study seeks to design an optimization-integrated deep learning model. The proposed research appears to improve upon the limitations of earlier studies and offers a more potent remedy for cloud resource management and security.

The Deep Linear Transition Network (DLTN), which exemplifies deep learning, and optimization methods such as the Adaptive Mongoose Optimization Algorithm (AMOA) are combined in the suggested methodology. This integration is promising since it uses neural networks to make smart decisions about how to allocate resources while streamlining the allocation procedure. Deep learning models are useful for resource allocation tasks because they can extract complex patterns from data, and optimization methods such as the AMOA can hone these selections for the best results. Resource allocation in cloud systems may become more effective as a result of this synergistic strategy.

Security is crucial in today’s environment of cloud computing. The Logic Overhead Security Protocol (LOSP)’s presence indicates a dedication to protecting the resource allocation procedure. Through the use of BAN logic to anticipate agreement logic, the LOSP improves the security posture of the cloud system. To guarantee that the LOSP efficiently performs its role in ensuring resource allocation, it is crucial to carry out extensive security assessments and take potential weaknesses into account.

The proposed methodology combines the power of deep learning, as represented by the Deep Linear Transition Network (DLTN), with optimization techniques such as the Adaptive Mongoose Optimization Algorithm (AMOA). This integration is promising because it harnesses the capabilities of neural networks to make informed resource allocation decisions while optimizing the allocation process. Deep learning models can learn intricate patterns from data, making them valuable in resource allocation tasks, while optimization algorithms such as the AMOA can fine-tune these decisions for optimal outcomes. This synergistic approach can potentially lead to more efficient resource allocation in cloud systems.

In the modern cloud computing landscape, security is of paramount importance. The inclusion of the Logic Overhead Security Protocol (LOSP) demonstrates a commitment to safeguarding the resource allocation process. By using BAN logic to predict agreement logic, the LOSP enhances the security posture of the cloud system. However, it is essential to conduct thorough security assessments and consider potential vulnerabilities that might arise in the implementation of the LOSP to ensure that it effectively fulfills its role in securing resource allocation.

While the proposed methodology offers significant promise, it is important to consider the potential complexities and computational overhead associated with deep learning and optimization algorithms. Deep learning models, especially when they involve convolutional neural networks, can be computationally intensive and may require substantial resources. Furthermore, the optimization process with the AMOA may add to the computational burden. Therefore, efficient implementation and scalability considerations are vital to ensuring real-world feasibility. The effectiveness of the proposed methodology ultimately hinges on rigorous performance evaluation and validation. It is essential to conduct comprehensive testing and benchmarking against existing resource allocation approaches. Metrics such as throughput, resource utilization, security robustness, and computational efficiency should be evaluated to determine whether the methodology offers tangible improvements over current practices.

The two primary objectives of the proposed methodology—creating a dynamic process for task scheduling and resource allocation and reducing the anticipated total time span—are commendable goals. The dynamic nature of cloud environments demands adaptable resource allocation strategies that can respond effectively to changing workloads. Additionally, minimizing make-up time is critical for ensuring that tasks are completed efficiently and within acceptable timeframes. The provided results and analyses offer valuable insights into the effectiveness of the proposed cloud resource allocation and security framework, DLTN-LOSP. The comparison of makespan, a crucial metric measuring the time needed to complete tasks, reveals that the proposed DLTN-LOSP model outperforms existing techniques by improving efficiency by approximately 30%. This reduction in time span signifies that the proposed model can efficiently allocate resources and complete tasks in a more timely manner, enhancing overall system performance.

The analysis of response time, which measures the duration from task arrival to completion, demonstrates that the proposed DLTN-LOSP model significantly minimizes response times compared to other algorithms. Reduced response times are desirable as they indicate quicker task completion, resulting in improved user satisfaction and system efficiency. The suggested DLTN-LOSP model’s many components are subject to a thorough execution time study, which is presented in Table 1. Understanding the computing requirements of the various framework components is made easier with the help of this breakdown. Real-time resource allocation in cloud systems requires efficient execution times, and the reported results indicate that the suggested approach performs well. The analysis of the resource consumption rate and task completion ratio demonstrates the suggested DLTN-LOSP model’s efficient resource allocation. When compared to competing algorithms, the model regularly obtains greater resource consumption rates, showing improved resource use. Additionally, it reaches a maximum task completion ratio, showing that the reward prediction process improves decisions regarding the allocation of resources, resulting in higher task completion rates. Figure 6 shows how the suggested DLTN-LOSP model successfully lowers overall energy usage, a crucial factor in cloud environments. The model’s resource allocation strategy is demonstrated by the power management module, which is used to achieve this decrease. The overall processing time analysis in Figure 7 shows how effective the suggested approach is in different test circumstances. The suggested DLTN-LOSP strategy regularly outperforms competing ones, demonstrating its capacity to effectively manage various scenarios. The suggested approach uses fewer virtual machines (VMs) for the same set of activities than existing methods, as shown in Figure 8. Due to the effectiveness of VM allocation, resources are used more effectively, and less hardware is needed, which eventually results in cost and resource savings. Figure 9 shows that, in comparison to alternative approaches, the suggested technique produces greater average resource utilization rates. This shows that, on average, resources are better used, improving overall resource efficiency. The analysis of load balancing performance in Figure 10 shows that the suggested DLTN-LOSP model makes load balancing much better by lowering the standard deviation of resource usage. By spreading out the resources more evenly, overloading and underutilization are avoided. The training and testing accuracy of the suggested model in comparison to other deep learning methods are shown in Figure 11 and Figure 12. The findings imply that the DLTN-LOSP model offers greater accuracy, highlighting the potency of the suggested resource allocation strategy.

The total utilization rate of the suggested and existing scheduling systems is shown in Figure 13 for various scenarios. Particularly when the number of VMs and tasks rises, the suggested DLTN-LOSP model displays greater utilization rates, indicating more effective resource utilization. The overall reward (cost) analysis is shown in Figure 14 and Figure 15, highlighting how the DLTN-LOSP strategy outperforms competing strategies. Lower total costs indicate higher resource use and more effective resource allocation. In summary, the full evaluation of the proposed DLTN-LOSP framework highlights its usefulness in enhancing several aspects of cloud resource allocation, including task completion time, response time, energy consumption, resource usage, and load balancing. These findings highlight the potential advantages of using this strategy in real-world cloud computing scenarios, where effective resource management is crucial for satisfying user expectations, maximizing resource usage, and lowering operational costs. To confirm the framework’s applicability to various cloud systems, it is crucial to stress that actual implementation and scalability considerations should be further investigated. The suggested methodology for deep-learning-based resource allocation and security in cloud systems offers a progressive solution to the problems with resource allocation in cloud environments. It aims to develop a dynamic and safe allocation process by integrating deep learning, optimization, and security features. However, for its potential benefits to be fully realized, successful implementation and validation are crucial. Additionally, the continued development of cloud technology will probably create new chances for innovation and improvement in this field.

### 5.1. Limitations

The following are some of the key limitations associated with this DLTN-LOSP model:The incorporation of deep learning (DLTN) and optimization algorithms can introduce significant computational complexity. Training deep neural networks and performing resource allocation optimization may require substantial computational resources, potentially limiting the model’s applicability in resource-constrained environments.The DLTN-LOSP model may demand substantial memory and processing power, particularly during training phases. This can lead to higher infrastructure costs for cloud service providers and may not be suitable for low-resource edge computing or IoT devices.Deep learning models such as the DLTN rely on large volumes of training data to make accurate predictions. Acquiring and maintaining such datasets can be challenging and may not always be feasible for all cloud service providers or organizations.Implementing and fine-tuning deep learning models and optimization algorithms require expertise in machine learning and cloud computing. Organizations without the necessary expertise may face challenges in deploying and managing the DLTN-LOSP model effectively.While the LOSP aims to enhance security, any security protocol is subject to potential vulnerabilities. The model’s security mechanisms must be regularly updated and rigorously tested to address emerging threats adequately.The model’s scalability may be limited in extremely large cloud environments with a high number of users and tasks. As the system scales, maintaining the model’s performance and efficiency may become increasingly challenging.Deep learning models such as the DLTN are trained on historical data and patterns. They may not generalize well to unseen or rapidly evolving scenarios. Adapting the model to changing cloud environments and user behaviors may require continuous retraining.Deep learning models are often considered “black boxes” because it can be challenging to interpret the rationale behind their decisions. This lack of interpretability may raise concerns in applications where transparency and accountability are essential.Training deep learning models requires access to sensitive data, potentially raising privacy concerns. Proper data anonymization and security measures must be in place to protect user information.Integrating the DLTN-LOSP model into existing cloud infrastructure may be complex and may require modifications to existing systems and processes. Compatibility issues could arise when integrating with legacy systems.Building, training, and maintaining deep learning models demand a workforce with specialized skills. The cost of hiring and retaining such expertise can be substantial.The accuracy and reliability of the model heavily depend on the quality of the training data. Inaccurate or biased data can lead to suboptimal resource allocation decisions.Cloud environments are highly dynamic, with resources and user demands continuously changing. The model may need constant updates and adjustments to adapt to these changes effectively.

In conclusion, while the DLTN-LOSP model offers promising solutions to resource allocation and security challenges in cloud computing, it is crucial to acknowledge and address its limitations. Organizations considering the adoption of this model should carefully assess their specific use cases, available resources, and the potential trade-offs associated with its implementation. Additionally, ongoing research and development efforts are necessary to overcome these limitations and refine the model’s performance in real-world cloud environments.

### 5.2. Future Work

Future works in the field of cloud resource allocation and security, building upon the DLTN-LOSP model, can explore various directions to address existing challenges and advance the state of cloud computing. Here are some potential areas for future research:Explore resource allocation models that seamlessly manage resources in hybrid cloud environments, combining on-premises infrastructure with public and private clouds. This research can address the challenges of workload migration, data placement, and security in hybrid setups.Enhance resource allocation models to consider multiple objectives simultaneously, such as cost minimization, energy efficiency, and quality of service (QoS) improvements. Multi-objective optimization techniques can help strike a balance between competing goals.Incorporate advanced threat detection and anomaly detection mechanisms within the DLTN-LOSP model to proactively identify and respond to security threats in real time. This can help prevent security breaches and data compromises.Explore the integration of blockchain technology to enhance the security aspects of resource allocation. The blockchain can provide a distributed and tamper-resistant ledger for tracking resource allocations and access control.Investigate how the advent of quantum computing may impact cloud resource allocation and security. Explore quantum-safe encryption methods and quantum-aware resource allocation algorithms.Continue research into energy-efficient resource allocation models that reduce the carbon footprint of cloud data centers. Green computing practices can align with sustainability goals.Develop autonomous resource allocation models that can make decisions without continuous human intervention. Autonomous systems can adapt to changing conditions and optimize resource usage dynamically.Explore the ethical implications of resource allocation and security decisions in cloud computing, especially in cases involving critical applications, healthcare data, and personal information.

Future research in these areas can contribute to the continued improvement of cloud resource allocation and security, addressing emerging challenges and ensuring the reliability, efficiency, and security of cloud services in an increasingly complex and dynamic computing landscape.

## 6. Conclusions

In this work, a novel resource allocation model incorporated with the security management model is developed for cloud systems. This work’s original contribution is the creation of a cutting-edge framework for distributing resources within the cloud system while ensuring security. Here, a dynamic process is developed for allocating resources and scheduling tasks. Then, the anticipated total manufacturing span is reduced while maximizing throughput through the use of optimal scheduling. After modeling the system, the SINR is calculated, and the empirical power model is built. The decision for resource allocation is then made using the DLTN algorithm after operations such as random action selection, convolutional feature estimation, model training, and loss function computation have been completed. As a result, the AMOA is used to forecast the reward for resource allocation and to compute the optimal beamforming solution. Following resource allocation, the allocation system’s security is ensured using the LOSP, where BAN logic is used to anticipate the agreement logic. During performance analysis, the results of the proposed DLTN-LOSP model are validated and compared using various parameters. The proposed model significantly reduces the makespan compared to existing approaches, resulting in an approximately 30% improvement in task completion time. It also minimizes response time, improves resource utilization, achieves a higher task completion ratio, reduces energy consumption, ensures quicker processing times, allocates fewer VMs, improves load balancing, exhibits higher training and testing accuracy, demonstrates scalability, and consistently achieves a better total reward (cost). Based on the results, it is evident that the proposed method provides an improved performance outcome over other approaches.

## Figures and Tables

**Figure 2 sensors-23-08448-f002:**
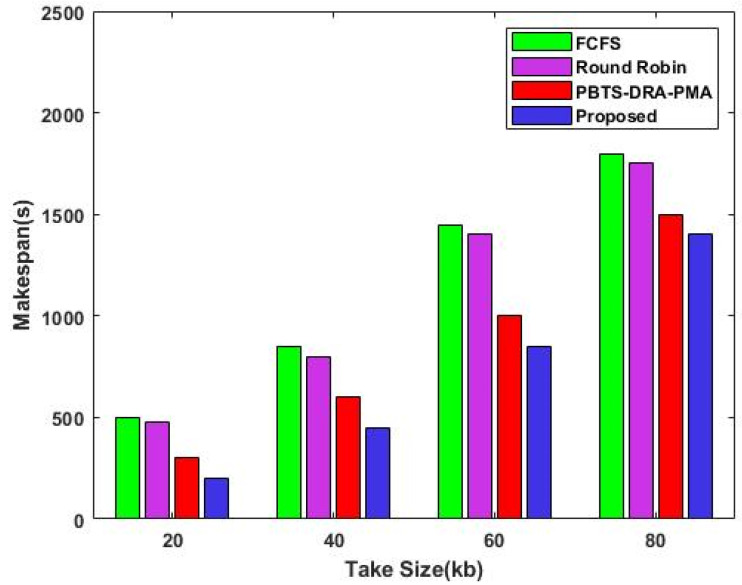
Makespan.

**Figure 3 sensors-23-08448-f003:**
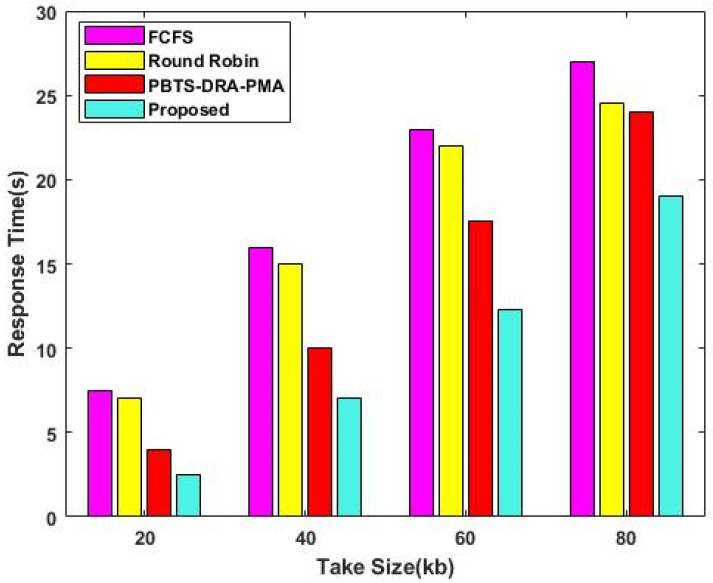
Response time.

**Figure 4 sensors-23-08448-f004:**
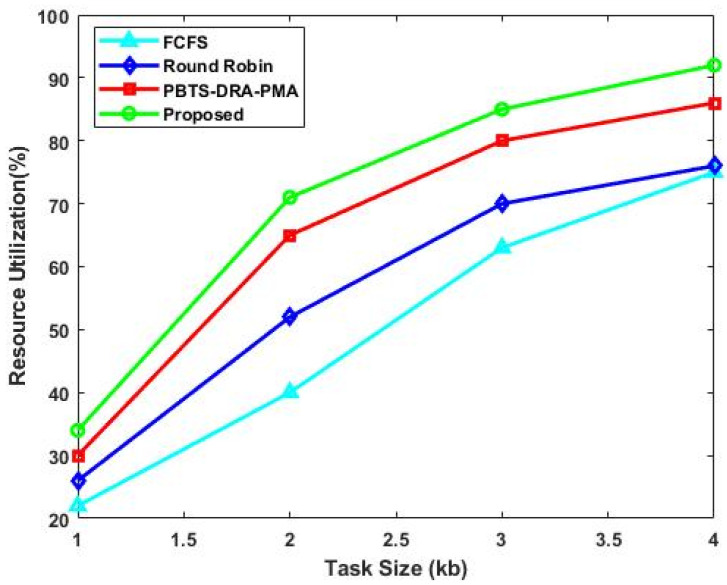
Resource utilization rate.

**Figure 5 sensors-23-08448-f005:**
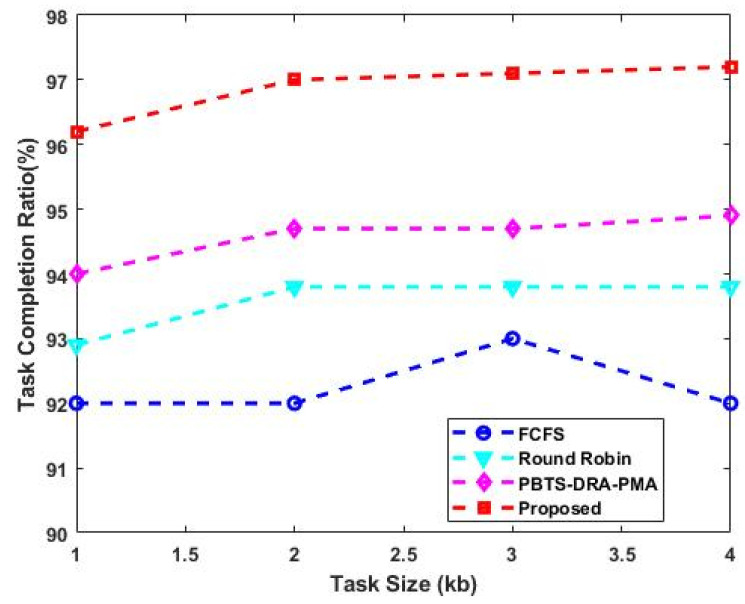
Task completion ratio.

**Figure 6 sensors-23-08448-f006:**
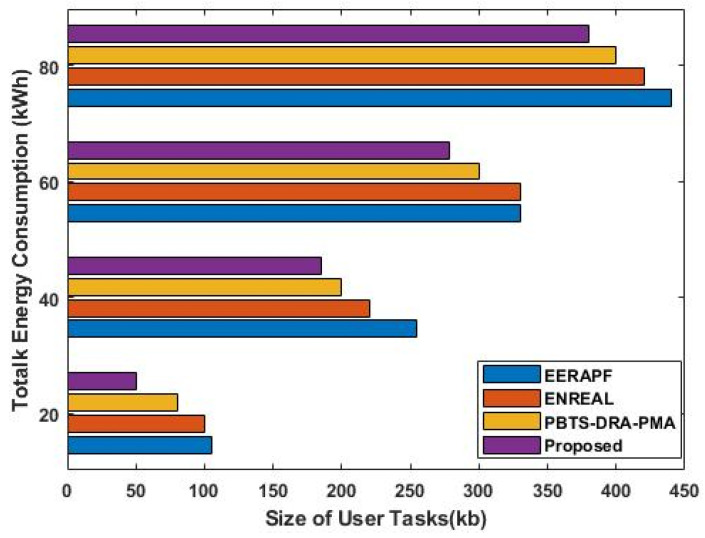
Total energy consumption.

**Figure 7 sensors-23-08448-f007:**
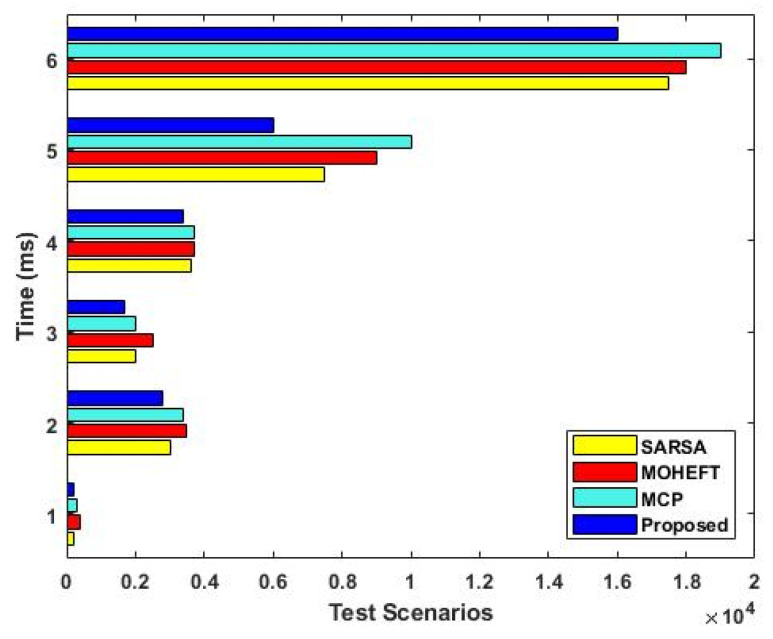
Overall processing time.

**Figure 8 sensors-23-08448-f008:**
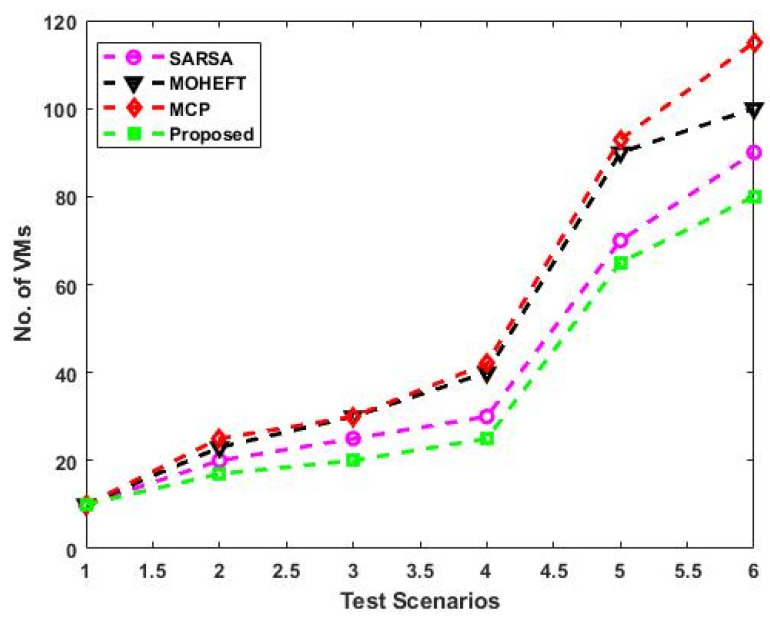
Number of allocated VMs with respect to different scenarios.

**Figure 9 sensors-23-08448-f009:**
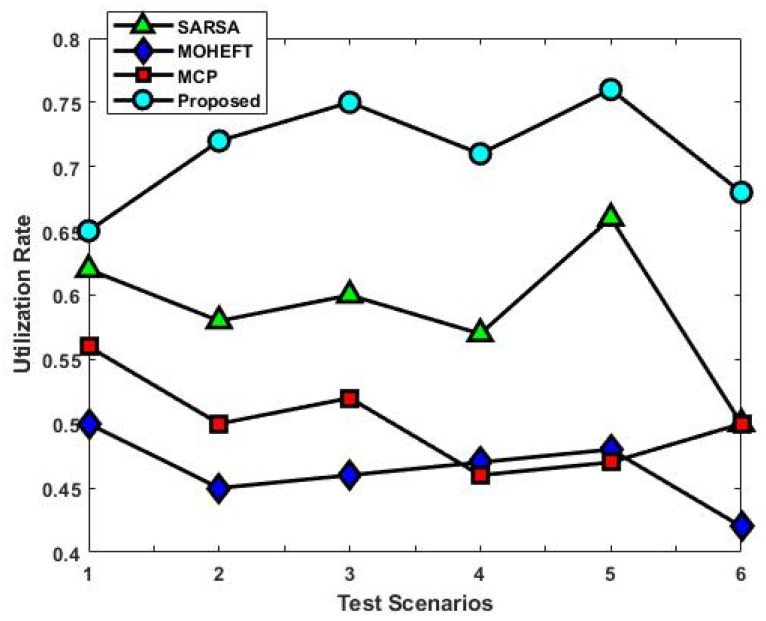
Utilization rate.

**Figure 10 sensors-23-08448-f010:**
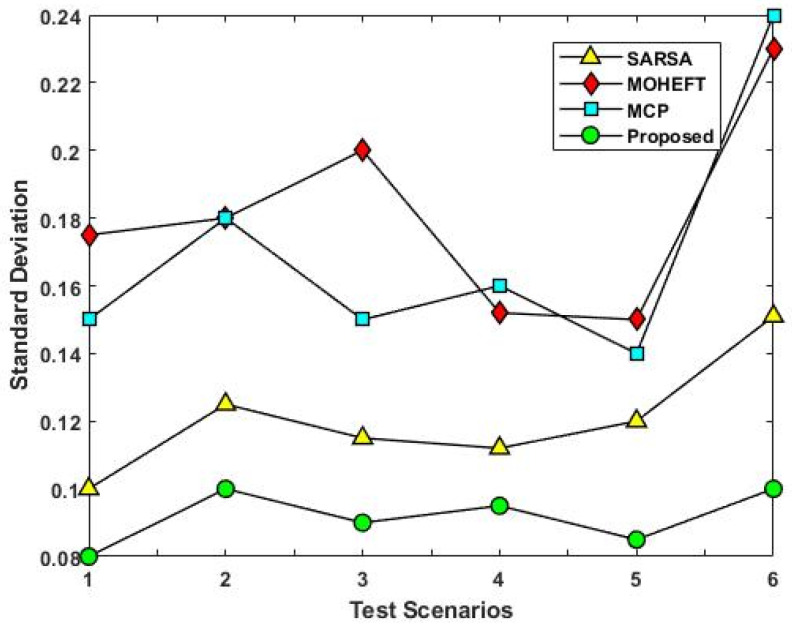
Standard deviation.

**Figure 11 sensors-23-08448-f011:**
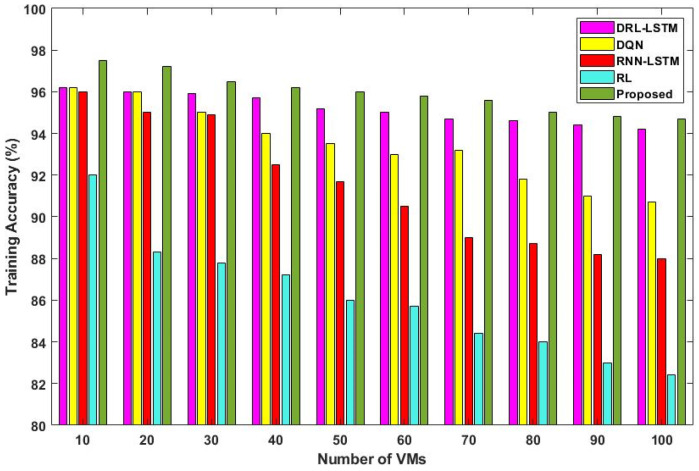
Training accuracy.

**Figure 12 sensors-23-08448-f012:**
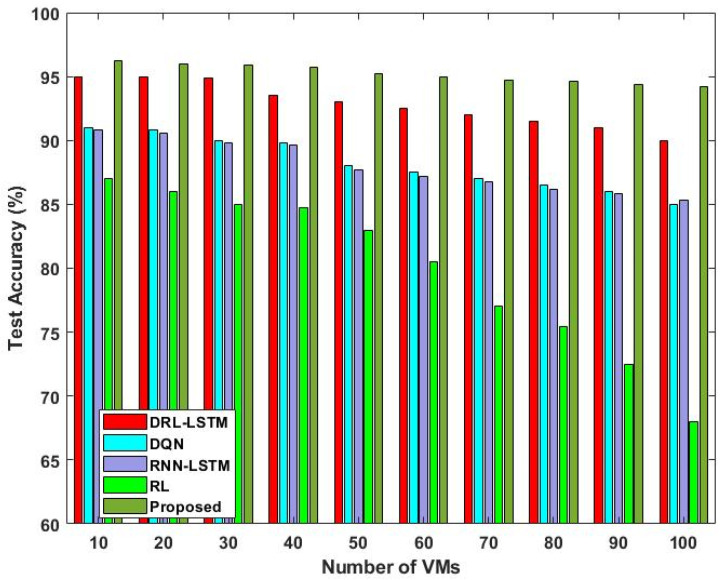
Testing accuracy.

**Figure 13 sensors-23-08448-f013:**
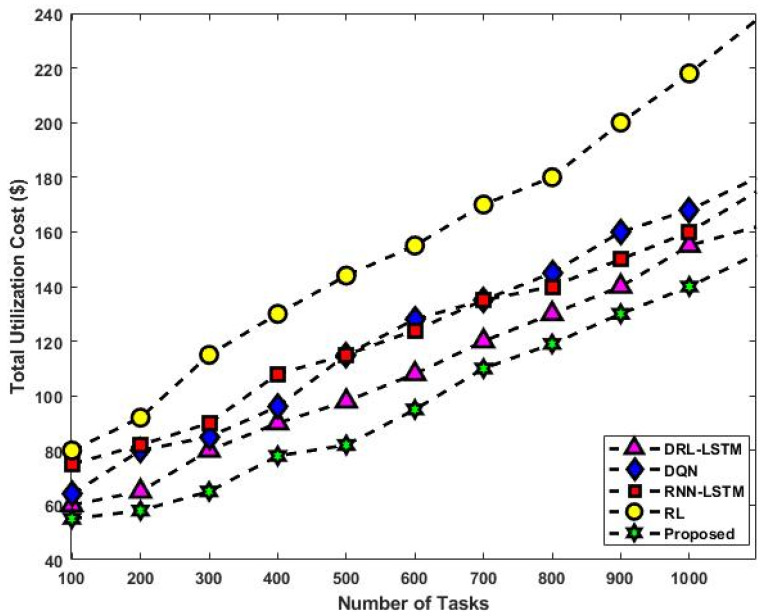
Total utilization rate.

**Figure 14 sensors-23-08448-f014:**
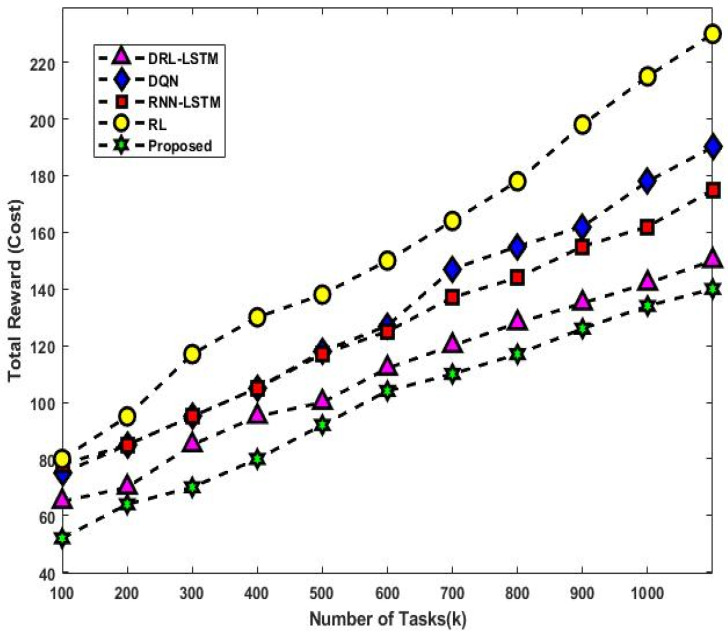
Total reward (cost) analysis.

**Figure 15 sensors-23-08448-f015:**
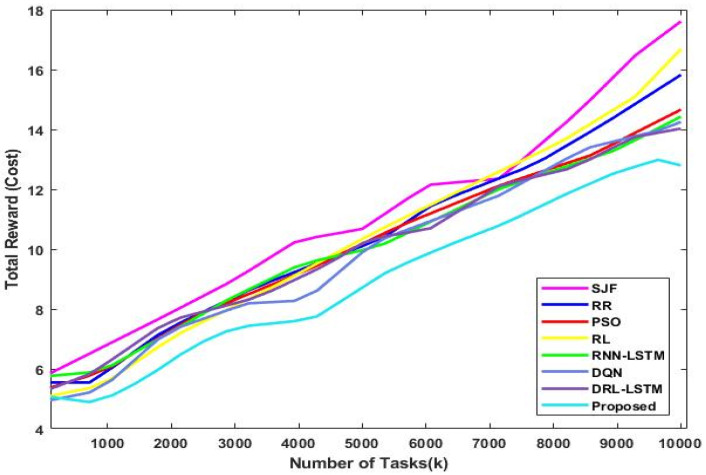
Total reward (cost) vs. number of tasks.

**Table 1 sensors-23-08448-t001:** A table summarizing the literature review, including the cited reference, methodology, results, and limitations of each work.

Cited Reference	Methodology	Results	Limitations
Hui et al. [24]	Lyapunov stability theory for resource allocation in mobile edge computing systems.	Ensured stability and safety of mobile edge systems through efficient resource allocation.	Not mentioned whether it considered security aspects.
Meng et al. [25]	Combined dynamic security-aware scheduling with Particle Swarm Optimization (PSO) for cloud resource allocation in industrial applications.	Optimized resource deployment to meet user requirements and considered security.	The performance of the suggested mechanism was not up to the mark.
Abid et al. [26]	Discussed various challenges in cloud resource allocation and categorized mechanisms into dynamic and AI-based approaches.	Highlighted challenges in predicting application needs, ensuring QoS, managing interruptions, and optimizing energy efficiency.	No specific methodology or results were mentioned.
Shukur et al. [27]	Explored methods and scheduling approaches for resource allocation in cloud computing via data center virtualization.	Demonstrated the capacity of various methods and scheduling algorithms to utilize shared cloud data center resources.	Specific results and methodologies not provided.
Du et al. [28]	Implemented a Deep Reinforcement Learning (DRL) algorithm incorporating LSTM and fully connected neural networks for optimal resource allocation in dynamic cloud systems.	Successfully accommodated more user requests with increased profit.	Faced problems of increased time consumption and overfitting.
Naha et al. [29]	Deployed a deadline-based dynamic resource allocation mechanism for fog–cloud environments with hierarchical resource ranking.	Reduced average processing time and cost.	Needed improvement in handling resource failures.
Afrin et al. [30]	Implemented a multi-objective resource allocation mechanism for a cloud-based smart factory application using NSGA-II algorithm.	Efficiently handled resource allocation with constraints.	High cost and makespan issues affecting overall framework performance.
Haji et al. [31]	Explored strategies for dynamic resource management to improve QoS parameters in cloud systems.	Focused on enhancing communication bandwidth, reducing traffic, and simplifying storage complexity.	No specific methodology or results mentioned.
Thein et al. [32]	Presented an energy-efficient resource allocation model for cloud systems using fuzzy logic and reinforcement learning.	Guaranteed SLAs, efficient resource utilization, and energy-efficient allocation.	Specific results not provided.
Praveenchandar et al. [33]	Deployed a dynamic resource allocation mechanism with a power management strategy for cloud systems.	Improved task scheduling and power management for energy efficiency.	No specific results mentioned.

**Table 2 sensors-23-08448-t002:** Experimental settings.

Components	Specification
Processor type	Intel, Pentium core i7, 3.778 GHz
OS	Windows 10, 64-bit OS
Hard Disk	1 TB
RAM	32 GB
Cloudlets	Length of tasks: 1500 to 3500 No. of tasks: 50 to 500
VM	Host: 4 Memory: 540
Physical machine (PM)	Bandwidth: 250,000 Storage: 500 GB
Hyperparameters considered	Loss function and learning rate

**Table 3 sensors-23-08448-t003:** Time analysis of the proposed DLTN-LOSP.

Parameters	Execution Time
Task monitoring and scheduling	18 min
Workload prediction	9 min
Agent connection	0.048 s
User response	0.010 s
Power management	1.198 s

## Data Availability

Not applicable.

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
