# Peer review of "DLTN-LOSP: A Novel Deep-Linear-Transition-Network-Based Resource Allocation Model with the Logic Overhead Security Protocol for Cloud Systems"

_sensors, 2023, doi:10.3390/s23208448_

Round 1
Reviewer 1 Report
Summary:
The manuscript presents a development a novel cloud resource allocation model incorporating security management which is conformed by:
- A Deep Linear Transition Network (DLTN) mechanism for effectively allocating resources to cloud systems.
- An Adaptive Mongoose Optimization Algorithm (AMOA) to compute the beam forming solution for reward prediction, which supports the process of resource allocation.
- A Logic Overhead Security Protocol (LOSP) to ensure secured resource management in the cloud system, where the BAN logic is used to predict the agreement logic.
The results presented provides an improved performance outcome over other approaches.
In the manuscript presented, the authors address a highly interesting and very current topic. In addiction, the manuscript is well organized and structured, nevertheless I consider that the authors should attend to certain minor issues in order to improve the good quality of work.
Comments for Authors:
- Minor grammatical errors have to be addressed.
- Even if the reader knows the subject, the authors should expand the first occurrence of any the acronyms or initialism, some of them are not expanded as examples BAN, VM, QoS, NN, IA-bases, etc.
- Page 12, row 476 authors mention: "LOSP employs Body Area Network (BAN) logic to predict agreement logic, enhancing the overall security of the cloud resource allocation system". In this case BAN logic is referring to Burrows–Abadi–Needham logic instead of Body Area Network logic.
- Page 13, row 476 figure 2 and 3. The change of colors in the series is confusing, you should use the same colors (same series, same colors).
- References to standards, de facto standard, equipment, components and software tools used must be included (i.e Azure dataset).
- In section Conclusion, the authors must highlight the results by referring quantitatively.
Author Response
Original Article Title: DLTN-LOSP: A Novel Deep Linear Transition Network-based Resource Allocation Model with Logic Overhead Security Protocol for Cloud Systems
To: Editor in Chief,
MDPI, Sensors
Re: Response to reviewers
Dear Editor,
Many thanks for insightful comments and suggestions of the referees. Thank you for allowing a resubmission of our manuscript, with an opportunity to address the reviewers’ comments.
We are uploading (a) our point-by-point response to the comments (below) (response to reviewers), (b) an updated manuscript with green, blue, and orange highlighting indicating changes, and (c) a clean updated manuscript without highlights (PDF main document).
By following reviewers’ comments, we made substantial modifications in our paper to improve its clarity, English and readability. In our revised paper, we represent the improved manuscript such as:
(1) Revised Abstract, (2) Revised Introduction, (3) Results section, (4) Discussions and Conclusion sections.
We have made the following modifications as desired by the reviewers:
Best regards,
Corresponding Author,
Dr. Qaisar Abbas (On behalf of authors),
Professor.

Reviewer 2 Report
Good work on the article. A few minor recommendations.
Please create a table of symbols for your mathematical equations.
Please align your mathematical equations to one side for easier reading.
Please increase the font in the result axes to be easier to visualize.
Please include Table 1 on a single page. Additionally, please rephrase the literature review to make it flow a bit better.
Lastly, please spread the equation spacing a little (if possible) to make it easier to visualize.
Author Response

(The authors gave the same response as above.)

Reviewer 3 Report
Please have a look at this reference - to include additional discussion on load balancing (at least in the literature review part): (https://link.springer.com/chapter/10.1007/978-3-642-16515-3_34)
A Task Scheduling Algorithm Based on Load Balancing in Cloud Computing
Web Information Systems and Mining, 2010, Volume 6318
ISBN : 978-3-642-16514-6
The list of problems (lines 85, 86) must include cybersecurity as it is your 3rd contribution to this study. Security concerns impact prioritization.
Deep Linear Transition Network (DLTN) for maximizing long-term reward is valid. The AMOA and its mathematical models is also valid. However, I could not understand your contribution in the LOSP. You have provided notations, definitions and use of the BAN logic, but how it relates to your work is not clear. (Either I have missed something or you need to clarify it further - from equations 19 to 24). I think you have done well with the previous two models; although discussion around security is important, I don't think LOSP is necessary for a narrowly defined study. You may want to redefine your study without LOSP or add more description to show its relationship with DLTN and AMOA. Please revisit lines around 488.
Your experiment setup, actual experiment and the results are very well described.
The limitations of your study (especially limited data and its costs) are also well described.
Some errors>>
118. Mobile edge or age?
276. Mangoose? Correct spelling in the figure
Please get the entire paper read by a professional proofreader. There are numerous errors in English that make the reading of the paper very difficult. Occasionally, it may even distort the meaning.
Author Response

(The authors gave the same response as above.)
